# Highly Selective Cleavage of TH2-Promoting Cytokines by the Human and the Mouse Mast Cell Tryptases, Indicating a Potent Negative Feedback Loop on TH2 Immunity

**DOI:** 10.3390/ijms20205147

**Published:** 2019-10-17

**Authors:** Zhirong Fu, Srinivas Akula, Michael Thorpe, Lars Hellman

**Affiliations:** Department of Cell and Molecular Biology, Uppsala University, SE-75124 Uppsala, Sweden; fuzhirong.zju@gmail.com (Z.F.); srinivas.akula@icm.uu.se (S.A.); getmeinahalfpipe@gmail.com (M.T.)

**Keywords:** mast cell, tryptase, chymase, serine protease, human chymase, cleavage specificity, cytokine, chemokine, TH2

## Abstract

Mast cells (MC) are resident tissue cells found primarily at the interphase between tissues and the environment. These evolutionary old cells store large amounts of proteases within cytoplasmic granules, and one of the most abundant of these proteases is tryptase. To look deeper into the question of their in vivo targets, we have analyzed the activity of the human MC tryptase on 69 different human cytokines and chemokines, and the activity of the mouse tryptase (mMCP-6) on 56 mouse cytokines and chemokines. These enzymes were found to be remarkably restrictive in their cleavage of these potential targets. Only five were efficiently cleaved by the human tryptase: TSLP, IL-21, MCP3, MIP-3b, and eotaxin. This strict specificity indicates a regulatory function of these proteases and not primarily as unspecific degrading enzymes. We recently showed that the human MC chymase also had a relatively strict specificity, indicating that both of these proteases have regulatory functions. One of the most interesting regulatory functions may involve controlling excessive TH2-mediated inflammation by cleaving several of the most important TH2-promoting inflammatory cytokines, including IL-18, IL-33, TSLP, IL-15, and IL-21, indicating a potent negative feedback loop on TH2 immunity.

## 1. Introduction

Mast cells are resident tissue cells of hematopoietic origin that primarily are found at the interphase between tissues and environment such as skin, intestinal mucosa, lungs, and close to blood vessels and nerves. These cells store massive amounts of immune mediators in cytoplasmic granules. A large fraction of the proteins stored in these granules are serine proteases, and all of these belong to the large family of trypsin/chymotrypsin-related serine proteases [1,2,3,4,5]. This protease family also includes several coagulation factors, complement factors, and the pancreatic digestive enzymes. The members of this family that are expressed by hematopoietic cells have been named hematopoietic serine proteases. They are primarily found in mast cells (MCs), neutrophils, natural killer (NK) cells, and cytotoxic T cells (Tc), where they are stored in their active forms in the granules for rapid release. Very high amounts of these proteases are found in these cells; the levels in mast cells can reach 35% of the total cellular protein [6]. hMCs express one chymotryptic enzyme, the MC chymase (HC), one enzyme with tryptic specificity, the tryptase and sometimes an enzyme, cathepsin G (hCG), which is otherwise primarily found in neutrophils. 

The various granule proteins of the hematopoietic cells have been shown to have a number of important immune functions including antibacterial, antiparasitic, general inflammatory, anti-inflammatory or apoptosis-inducing activity. For example, the mouse counterpart of the HC, mouse mast cell protease-4 (mMCP-4) has been shown to be very potent in inactivating several snake, bee, and scorpion toxins, indicating an important role in defense against potentially life threatening toxins, a function that is probably evolutionary very old [7]. Recently, some of the hematopoietic serine proteases have been shown to cleave, and thereby inactivate, various cytokines and chemokines. For example, mMCP-4 cleaves and inactivates tumor necrosis factor-alpha (TNF-α), thereby limiting the inflammatory response [8]. The HC also cleaves and activates IL-1β and IL-18, as well as cleaving the region just outside of the membrane of membrane bound stem cell factor (SCF), which releases it from the cell [9,10,11]. The release of SCF from the cell surface makes it able to move more freely in the tissue, which may then be of importance for attracting progenitor cells, primarily MC progenitors, from the blood. In addition, HC has been found to cleave two chemotactic substances: active chemerin and eotaxin-3 (CCL26) [12,13]. The degradation of IL-33, an IL-1-related cytokine, by mMCP-4 and HC indicates that they potentially have a role in limiting inflammation [14,15]. Another of the hMC enzymes, the tryptase, has also been shown to efficiently degrade the chemokine eotaxin [16]. Numerous examples on the role of these enzymes in the degradation of inflammatory mediators have been described, indicating that this may be one of the important functions of these enzymes. In order to study the potential roles of these in limiting inflammation by cleaving cytokines and chemokines in more general terms, we recently performed an extensive analysis of the cleavage of 51 different human cytokines and chemokines by the HC and hCG. The results showed a remarkable selectivity for both enzymes but primarily for the HC. Only significant cleavage in 3–4 of the 51 studied cytokines and chemokines for this enzyme was detected [17]. The cleavage by both enzymes of two IL-1-related cytokines, IL-18 and IL-33, which act as alarmins, indicated a role of these enzymes in limiting excessive inflammation. As a continuation of these studies, here we present a similar study of the human and the mouse mast cell tryptases. The tryptase found in human MCs is a homo or heterotetramer of four closely related proteases the α, β1, β2, and β3 tryptases [1]. Interestingly the formation of a heteroteramer between the proteolytically inactive α-tryptase and one of the active β-tryptases changes both the stability and specificity of the β-tryptase [18]. The mouse counterpart is named mouse mast cell proteases 6 (mMCP-6) [19]. Interestingly, the active sites of these four subunits are positioned in the center of the tetramer, making them less accessible for larger substrates [20].

The human tryptase was found to be even more specific than the HC and we could only observe cleavage of five out of 69 different cytokines and chemokines. The cytokines that were efficiently degraded were TSLP and IL-21 and the chemokines MCP3, MIP-1b, and eotaxin. Interestingly, when combining the cleavage of the two of the major proteolytic enzymes of hMCs, the tryptase and the chymase, we can now see that together they cleave three of the most potent TH2 promoting cytokines, IL-18, TSLP, and IL-33. Interestingly, also IL-15 and IL-21, which both are efficiently cleaved by HC and the tryptase, respectively, have been indicated in either promoting TH2 immunity or inhibiting TH1 immunity [21,22,23]. This indicates that one major function of the MC proteases is to limit excessive TH2 driven inflammation. They may act together as a negative feedback loop by cleaving and thereby inactivating early TH2 promoting inflammatory cytokines.

## 2. Results

### 2.1. Analysis of the Purity and Activity of the Recombinant Human and Mouse Mast Cell Tryptases

To determine the activity and purity of the recombinant human tryptase, the enzyme was dissolved in assay buffer and a sample of approximately 2 µg was separated on a 4–12% SDS-PAGE gel (Figure 1a). The figure showed several bands that most likely originate from heterogenous glycosylation (Figure 1a). Expression in fungal expression systems like the *Pichia pastoris* often generates heterogenous glycosylation. However, deglycosylation of proteins produced in this system, using the same purification strategy as this commercial enzyme have shown high purity and that all the diverse bands observed on gels originate from differently glycosylated tryptase [24].

The activity of this recombinant tryptase was then tested on three different chromogenic substrates and the enzyme was found to be highly active against all three of these low molecular weight substrates, which shows the enzyme has a high proteolytic activity (Figure 1b). The mouse mast cell tryptase, mMCP-6 was produced in the human cell line HEK293-EBNA, and after purification on Ni^+2^ chelating IMAC columns activated by cleavage by enterokinase, lowering the pH to 6.0, and adding heparin, as previously described (Figure 1c) [19].

The activities of both enzymes were analyzed against three chromogenic tryptase substrates (Figure 1b,d).

### 2.2. Analysis of Cleavage Sensitivity against a Panel of 69 Cytokines and Chemokines by the Recombinant Human Tryptase

The cleavage activity on 69 different recombinant human cytokines and chemokines by the recombinant human tryptase was analyzed in 11 µL cleavage reactions with approximately 1.2 µg of cytokine and chemokine and 13 ng of the tryptase (Figure 2). To confirm the initial results, the experiment was repeated under the same conditions as previously described. The results were the same as the first experiment (data not shown). In both experiments the enzyme to target ratio was the same, approximately 1:92.

Most of the 69 cytokines and chemokines analyzed were not cleaved by this enzyme. Of the cytokines analyzed, only two were efficiently cleaved, TSLP and IL-21, and we could only observe efficient cleavage of three chemokines, MCP3, MIP-3b, and eotaxin (Figure 2). A minor N or C terminal trimming of IFN-γ, IL-8, and IP-10, and a minor degrading effect on IL-7, SDF-1β, and CTGF could also be detected (Figure 2). By using a three-fold increase in the amount of enzyme, we observed a more pronounced degrading effect on CTGF, and also cleavage of IGF-1, IGF-2, and FGF-1, indicating that some more targets appear with increasing enzyme to target ratio. However, generally, the tryptase was remarkably restrictive in its cleavage of this large panel of cytokines and chemokines (Figure 2a,b). Interestingly, where cleavage occurred, it appeared as if it degraded the target almost completely as only very faint bands for fragments could be detected for three of the most sensitive targets, TSLP, IL-21, eotaxin, and MIP-3b and there were no traces of MCP3 after cleavage (Figure 2).

### 2.3. Analysis of Cleavage Sensitivity against a Panel of 56 Cytokines and Chemokines by the Recombinant Mouse Tryptase, mMCP-6

The cleavage activity on 56 different recombinant mouse cytokines and chemokines by the recombinant mouse tryptase, mMCP-6, was performed (two times with identical result) under the same conditions as described for the human tryptase (Figure 3).

Most of the 56 cytokines and chemokines analyzed were not cleaved by this enzyme. Of the cytokines and growth factors analyzed, only eight were efficiently cleaved, IL-13, IL-9, IL-21, IL-33, VEGF-A, PDGF-B, IL-17C, and IGF-1 (Figure 3), and we could only observe efficient cleavage of five chemokines, IP-10, MCP-1, MIP-3a, MIP-3b, and eotaxin (Figure 3). A minor N or C terminal trimming of IFN-γ, IL-6, IL-11, and IL-17F, and a minor degrading effect on IL-7 and SDF-1α could also be detected (Figure 3).

### 2.4. Analysis of Structural Similarities between the Three Efficiently Cleaved Cytokines and Chemokines

To try to understand why a few of the cytokines and chemokines were efficiently cleaved and why the remaining majority were almost totally unaffected by this enzyme, we wanted to see if there were any structural characteristics in common between the three cleaved cytokines and chemokines. After analyzing the sequence, all three contained positively charged patches (Figure 4). Human TSLP was probably the most extreme in this case, with seven basic amino acids in a row (KKRRKRK) (Figure 4). Interestingly this sequence is lacking in both rat and mouse TSLP but present in human and dog TSLP (Figure 4). In order to study if this difference affected the efficiency in cleavage by the human tryptase, we tested cleavage of both human and mouse TSLP. Human TSLP was very efficiently cleaved and was totally degraded by 13 ng of tryptase, whereas mouse TSLP was not cleaved at all, even when using 10 times higher enzyme concentration, 136 ng, which clearly indicates that this positively charged patch is of importance for the cleavage (Figure 5a). This sequence (KKRRKRK) is not found in any other cytokine of chemokine in the entire human proteome. All other human cytokines and chemokines analyzed in this study also lacked a positive patch except VEGF-A, PDGF-A and B. A C-terminal patch of positively charged amino acids were also seen in IFN-γ. Interestingly IFN-γ is trimmed most likely in the C-terminal (Figure 2). However, no cleavage by the human tryptase of VEGF-A, PDGF-A and B was observed, indicating that also higher order structure of the protein is of importance. Interestingly mMCP-6 did cleave mouse VEGF-A, PDGF-A and B, indicating that even minor differences in the structure of the protease or of the target may affect this selectivity. It should be noted that these three cytokines are highly homologous and almost identical in the regions of interest, possibly except VEGF-A where the mouse protein has a big C-terminal deletion, which did not affect the positively charged region. The deletion may possibly enhance the accessibility of this charged region, making it a better target for mMCP-6. Higher order structures most likely play an important role in the selectivity, if the positively charged region is exposed or not on the surface of the protein. Cleavage may also occur only if the protein has an unstructured region accessibly for cleavage by the tryptase. Two structural requirements may therefore have to be fulfilled, a positive patch for targeting and a loose structure to allow targeting by the enzymes.

### 2.5. Analysis of the Effect of Spermine on Tryptase Activity

As indicated from previous sections, a high positive charge, as exemplified by the KKRRKRK sequence in human TSLP, seems to be an important factor for the selectivity of the tryptase. To look deeper into this selectivity, we decided to analyze the effect on cleavage by a potential low molecular weight competitor. Spermine appeared as a good candidate. It has a molecular weight of only 202.34 g/mol and at physiological pH all of its four amino groups are charged. The spermine solutions were made fresh from powder just before use and pH adjusted to pH 7.2. Spermine and tryptase were first mixed in the reaction buffer and left to equalize for a few minutes before starting the experiment by addition of the chromogenic substrate. No effect on the cleavage of the chromogenic substrate was seen at concentrations of spermine from 0.1 to 3 mM, indicating that spermine has no effect on the cleavage on low molecular weight substrates (Figure 5b). However, when we analyzed the cleavage of human TSLP the effect of spermine was quite dramatic (Figure 5c). Already at a concentration of 0.5 mM, the cleavage was reduced by approximately 20%, at 1 mM the cleavage was inhibited by approximately 50%, and at 3 mM almost totally inhibited, which shows a strong inhibitory effect of spermine on the cleavage of the positively charged target human TSLP by the human tryptase (Figure 5c).

### 2.6. Cleavage of other Substrates by Human Tryptase

Several proteins have previously been shown to be trimmed or degraded by human tryptase including fibrinogen and fibronectin [26,27]. To confirm these results and look at potential mechanisms, we performed a cleavage reaction with purified human fibrinogen and human fibronectin (Figure 5d). We could not detect any cleavage of fibronectin and only minor trimming of fibrinogen (Figure 5d). It is known that the N terminal tails of fibrinogen α and β chains are relatively unstructured and open for cleavage by thrombin (Figure 5e) [25]. The cleavage of the ends of the α and β chains of fibrinogen by thrombin results in the polymerization of fibrinogen into fibrin clots. It is highly possible that these loose unstructured ends make them accessible for cleavage also by the human tryptase, whereas the more tightly structured remaining parts of fibrinogen are more difficult for the tryptase to access.

In marked contrast to the tryptase, both human and opossum chymases did cleave both fibronectin and fibrinogen quite efficiently (Figure 5d). Furthermore, the cleavage patterns generated were quite similar indicating a conservation in target specificity over more than 150 million years of mammalian evolution.

### 2.7. Cleavage Selectivity of Wild Type (WT) and Mutant Human Tryptase

By a detailed analysis of the amino acid sequence of the human tryptase we observed a region just C-terminally of the Asp of the active site of the enzyme in a panel of tryptases. All had negatively charged amino acids, whereas a panel of chymases in the same region was positively charged (Figure 6a). To test if this region was involved in guiding positively charged substrates into the tetramer for efficient cleavage, we produced wt and mutant human tryptase involving these two residues (Figure 6a,b). We exchanged two glutamic acid residues of the tryptase into lysines, the amino acids found in these positions in the chymases (Figure 6a). The wt and mutant tryptases were produced in the human cell line HEK-293 EBNA and tested for their activity against three chromogenic substrates and against recombinant human TSLP (Figure 6c,d). To our surprise, both enzymes were equally active against all of these substrates, showing that this region of the protease has no effect on the selectivity of the tryptase (Figure 6c,d).

## 3. Discussion

In a recent study, we have shown that both HC and hCG show a relatively selective cleavage of a panel of human cytokines and chemokines [17]. This finding contradicted the previously dominating view of the hematopoietic serine proteases as being relatively unspecific and cleaving almost any substrate if allowed to do so for extended periods of time. In order to broaden this analysis, here we have studied another dominating serine protease of human MCs, namely the tryptase. Here, we can see that this enzyme is even more restrictive. For the human tryptase, we could observe efficient cleavage of only five cytokines and chemokines out of the 69 tested (Figure 2). All of the five that were efficiently cleaved had one interesting characteristic in common, the presence of one or several patches of highly positively charged residues (Figure 4). If that patch was not present, as tested by cleavage of mouse TSLP, this protein was totally resistant to cleavage even with 10 times more enzyme (Figure 5a). Mouse TSLP is similar in its overall structure to its human counterpart but lacks this positive patch. Human TSLP was also very efficiently cleaved and almost no traces of the target could be seen on the SDS-PAGE gel, indicating that if the substrate enters the tetramer it becomes fully degraded (Figure 1e and Figure 2). This shows clear similarities with the general cytoplasmic proteasome, which primarily cleaves polyubiquinated substrates [28,29]. The tryptase also show similarities to the coagulation protease thrombin, where positively charged patches on the enzyme, so called exosites, attract negatively charged regions on the substrate and thereby bring the target close to the active site [30].

The question is what region on the tryptase acts as an exosite? One very interesting candidate is heparin, which is not part of the enzyme itself but is attached to the tryptase, acting as a stabilizer and activator of the enzyme. When producing the tryptase as a recombinant protein, the enzyme is inactive before adding heparin [19]. Heparin then assists the tetramerization of the enzyme. At neutral pH, tetramerization is most likely essential for activity and here heparin is part of this activation step [19]. Heparin is the most negatively charged molecule in the human body and is thereby a likely candidate for the selectivity of tryptase for positively charged substrates [31,32]. Interestingly, a similar effect has previously been observed for the other dominating proteolytic enzyme of human connective tissue MCs, the chymase. Chymase needs heparin for efficient targeting of several targets including fibronectin, thrombin, and plasmin [33,34]. The mechanism is probably very similar to the proposed effect on tryptase, where heparin acts as a binding surface to attract the target and increase the target concentration close to the active site. These findings add additional support for the very important role of heparin and other highly negatively charged proteoglycans such as different chondroitin sulfates in MC biology [31,32].

In order to test the importance of the positive patch for the target selectivity of the human tryptase, we analyzed the effect of a spermine on the cleavage of chromogenic substrates and of human TSLP. We observed no effect by spermine on the cleavage of a low molecular weight chromogenic substrate, which contains no positive patch; however, a potent effect by spermine on the cleavage of human TSLP was observed (Figure 5b,c). This latter potent inhibitory effect by spermine on the cleavage of a positively charged substrate indicates that the interaction between a positively charged substrate and the enzyme, with its attached negatively charged heparin, is inhibited by the presence of a positively charged molecule like spermine, which supports the role of a positive patch on the substrate for the targeting by the tryptase.

Upon detailed analysis of a panel of tryptases and a panel of MC chymases, we also observed a striking difference. In a region just C-terminally of the asparagine of the catalytic triad, there were several positively charged residues in all the chymases, whereas in all tryptases this region was negatively charged (Figure 6a). To test the possibility that this region was responsible for guiding positively charged substrates into the mouth of the tryptase, we produced wt and mutant human tryptase, where the mutant had two of the negatively charged amino acids of the tryptases exchanged for the corresponding positively charged residues of the chymases (Figure 6a). Interestingly, both wt and mutant enzymes were equally active against both chromogenic substrates and on human TSLP, indicating that these residues had nothing to do with the selectivity for positively charged substrates (Figure 6c,d). This adds additional support for the idea that it is not the enzyme itself but heparin that is the prime reason for this selectivity, as there are no other regions with strong negative charge on the enzyme.

In addition to targets with a positive patch, proteins that have unstructured regions and are therefore more accessible to enter the tetramer may still be cleaved, as seen for fibrinogen, where only the relatively unstructured ends were cleaved (Figure 5d,e).

Many of the potential targets identified for the human tryptase are actually also small peptide hormones, like vasoactive intestinal peptide (VIP) and several airway neuropeptides [1,35,36]. These may more easily enter the tetramer (Figure 1e).

When we look at the panel of cytokines cleaved by the two MC enzymes, one striking feature appears; these two enzymes together cleave several of the most potent TH2 promoting cytokines, including IL-18, IL-33, and TSLP (Table 1). This finding indicates that these MC enzymes have a prominent role in dampening a TH2 driven inflammation, and inflammation at least partly is initiated by the same cells, namely MCs and basophils. This points towards a negative feedback loop by these enzymes on a TH2 dependent inflammatory response.

In a parallel project, we have been working on the development of therapeutic vaccines against allergy, and there focused on a few early TH2-inducing cytokines as potential targets for an allergy vaccine [37,38,39]. Interestingly, both of these projects have resulted in the identification of the same three cytokines, TSLP, IL-18, and IL-33 (Figure 2, Figure 3 and Figure 7). The interest for these early TH2-inducing cytokines from completely different origins does, in our mind, strongly favor a role of the MC proteases to limit excessive TH2-driven inflammation by cleavage, and thereby inactivation, of several of these early TH2-promoting inflammatory cytokines (Figure 7). The MC proteases may thereby act as a negative feedback loop to regulate the inflammation partly initiated by the same cell.

We also observed efficient cleavage of IL-15 by the HC and of IL-21, MCP3, and eotaxin by the tryptase, indicating an even broader anti-inflammatory effect of these two MC enzymes, when they act together. IL-15 is an important cytokine for NK cell activation, and cleavage of this cytokine by the chymase can thereby also have a dampening effect on NK cell activity [43]. However, and interestingly, Il-15 has been shown to enhance TH2 immunity and blocking IL-15 has been shown to prevent the induction of allergen-specific T cells and allergic inflammation in vivo, indicating that IL-15 can also act as a potent TH2-promoting cytokine [21,22]. IL-21, which was also efficiently cleaved by the tryptase, has been shown to downregulate TH1-mediated immunity, indicating that it may act as a TH2 cytokine [23]. HC and the human tryptase thereby seem to act together in a potential negative feedback loop on TH2-mediated immunity by cleavage of TSLP, IL18, IL-33, IL-15, and IL-21. The potent cleavage of two of the most important chemokines for eosinophil and basophil influx into an area of inflammation, eotaxin and MCP3, further supports the role of these two enzymes in controlling excessive TH2 immunity.

Several studies on the potential role of these enzymes or cytokines in vivo have been performed, which gives strong support the role of these mast cell proteases in controlling excessive TH2 immunity. One very important study is a study of the mouse counterpart of the human mast cell chymase, mMCP-4. Knock out of mMCP-4 shows a strong effect on both sensitization and IgE levels [14]. This study gives a strong indication that the cleavage of IL-33 and potentially also IL-18 and IL-15 by mMCP-4 is important for the control of excessive TH2-mediated immunity in mice and most likely also in humans. Two important studies on the effect of blocking IL-15 in vivo have also been published [21,22]. Blocking IL-15 in vivo has been shown to prevent the induction of allergen-specific T cells and allergic inflammation, which further substantiates the effect of reducing the levels of these cytokines on allergic sensitization. Also, IL-21 has been shown in vivo to act as a TH2 cytokine by suppressing the induction of a TH1 response [23]. There are a number of important in vivo studies that support the role of these enzymes in regulating TH2 immunity by cleavage and thereby reduction in the in vivo levels of these selected cytokines and thereby on allergic sensitization. What also is remarkable is the high selectivity of the HC and human tryptase on these TH2 cytokines and almost no effect on the majority of all the other cytokines or chemokines tested.

Interestingly, concerning the roles of MC proteases in cytokine regulation, is the fact that human TSLP was degraded but not mouse TSLP, indicating different functions between different species. A similar phenomenon was seen in our previous analysis of the HC. Here, the human enzyme was seen not to cleave the inflammatory cytokine TNF-α, which is in marked contrast to its mouse counterpart mMCP-4, which efficiently cleaves mouse TNF-α [8,17]. This clearly indicates that the targets may vary quite extensively between relatively closely related species, but that the effects may still be quite similar, in this case an anti-inflammatory effect. The same was seen for the mouse and human tryptases. Both efficiently cleaved IL-21, MIP-3B, and eotaxin; however, they differ in the cleavage of two other TH2 cytokines. The human tryptase cleaves TSLP, whereas mMCP-6, the mouse tryptase, instead cleaves IL-13 (Figure 2 and Figure 3). Both of these cytokines are important TH2 cytokines whose effect may be similar even if some of the targets may differ.

In summary, our analysis of this broad panel of 69 different active human cytokines and chemokines for their sensitivity to cleavage by the human MC tryptase showed a remarkably restrictive cleavage. Only five out of 69 were efficiently cleaved. This is together with the knowledge that the HC has previously been shown to be highly selective in its cleavage of cytokines and chemokines, and together they act on a few inflammatory cytokines including several of the most prominent TH2-inducing cytokines, IL-18, IL-33, TSLP, IL-21, and IL-15. These findings concerning the potent activity on TH2-inducing cytokines by these two prominent MC enzymes needs to be taken into careful consideration when studying the effects of protease inhibitors targeting chymase or tryptase in allergy treatment. Interestingly, the mouse counterpart of the human tryptase, mMCP-6, was almost as restrictive in its cleavage. It also cleaved almost all the cytokines and chemokines cleaved by the human enzyme, except TSLP. However, to compensate for this loss in activity on TSLP, mMCP-6 instead cleaves two other important TH2 cytokines, IL-13 and IL-9.

## 4. Materials and Methods

### 4.1. Enzymes and other Reagents

The recombinant human tryptase was purchased from Promega Biotech (Madison, WI, USA). This enzyme was expressed in a fungal (*Pichia pastoris*) expression system. Purified human fibrinogen (natural human) and fibronectin (from human plasma) were purchased from Abcam (Cambridge, UK) and Gibco-Life Technologies (Frederick MD, USA), respectively. Spermine was purchased from Sigma Aldrich (S3236) (Saint Louis, MO, USA). mMCP-6, the human and opossum chymases, and the wt and mutant human tryptase were produced in the mammalian cell line HEK293-EBNA with the vector pCEP-Pu2 according to previously published procedures [44,45,46]. The coding regions for full-length enzymes containing an N-terminal 6-histidine purification tag were order as designer genes from Genscript (Piscataway, NJ, USA), and the proteins were purified from conditioned media from transfected HEK293-EBNA cells on Ni-chelating IMAC agarose (Qiagen, Hilden, Germany).

#### Recombinant Human and Mouse Cytokines and Chemokines

Sixty-six recombinant human (rh) cytokines and chemokines, and 56 mouse cytokines and chemokines were purchased from Immuno Tools (Friesoythe, Germany). rhIL-25 from R&D systems (Abingdon, UK) rhIL-18 from MBL (MBL International, Woburn, MA, USA), rhIL-33 from GIBCO (Invitrogen Corporation, Camarillo, CA, USA).

### 4.2. Analysis of the Sensitivity to Cleavage by the Recombinant Human and Mouse Tryptase

The cytokines and chemokines were dissolved in PBS or sterile water, according to the recommendations of the supplier (Immuno Tools, Friesoythe, Germany), to get an approximate concentration of 0.13 µg/µL. Subsequently, 9 µL (~1.2 µg) of the cytokine was mixed with 2 µL of the recombinant human tryptase (~13 ng) or in-house produced mMCP-6 and incubated for 2.5 h at 37 °C. Two µL of PBS was used as control. The cleavage was performed in 1× PBS at pH 7.3. After incubation, the reactions were stopped with the addition of 3 µL of 4x sample buffer. Half a microliter of β-mercaptoethanol was then added to each sample followed by heating for 7 min at 85 °C. The reaction mixtures were then analyzed on 4–12% precast SDS-PAGE gels (Novex, Invitrogen, Camarillo, CA, USA). To visualize the proteins, the gels were stained overnight in colloidal Coomassie staining solution and destained with 25% (*v*/*v*) methanol in ddH_2_O for 4 h [47]. The analysis was completely repeated and the result was identical between the two independent experiments.

### 4.3. Analysis of the Cleavage of Human Fibrinogen and Fibronectin

Human fibronectin: 6 µg per lane was cleaved with three different enzymes for 30 min at 37 °C, with the following amounts of enzyme: human tryptase 136 ng, HC 272 ng and the opossum chymase 500 ng. Human fibrinogen: 4 µg per lane was cleaved with three different enzymes for 30 min at 37 °C, with the following amounts of enzyme, human tryptase 136 ng, HC 136 ng, and opossum chymase 500 ng. After incubation, the reactions were stopped with the addition of 3 µL of 4× sample buffer. Half a microliter of β-mercaptoethanol was then added to each sample followed by heating for 7 min at 85 °C. The reaction mixtures were then analyzed on 4–12% precast SDS-PAGE gels (Novex, Invitrogen, Camarillo, CA, USA). To visualize the proteins, the gels were stained overnight in colloidal Coomassie staining solution and destained with 25% (*v*/*v*) methanol in ddH_2_O for 4 h [47].

## Figures and Tables

**Figure 1 ijms-20-05147-f001:**
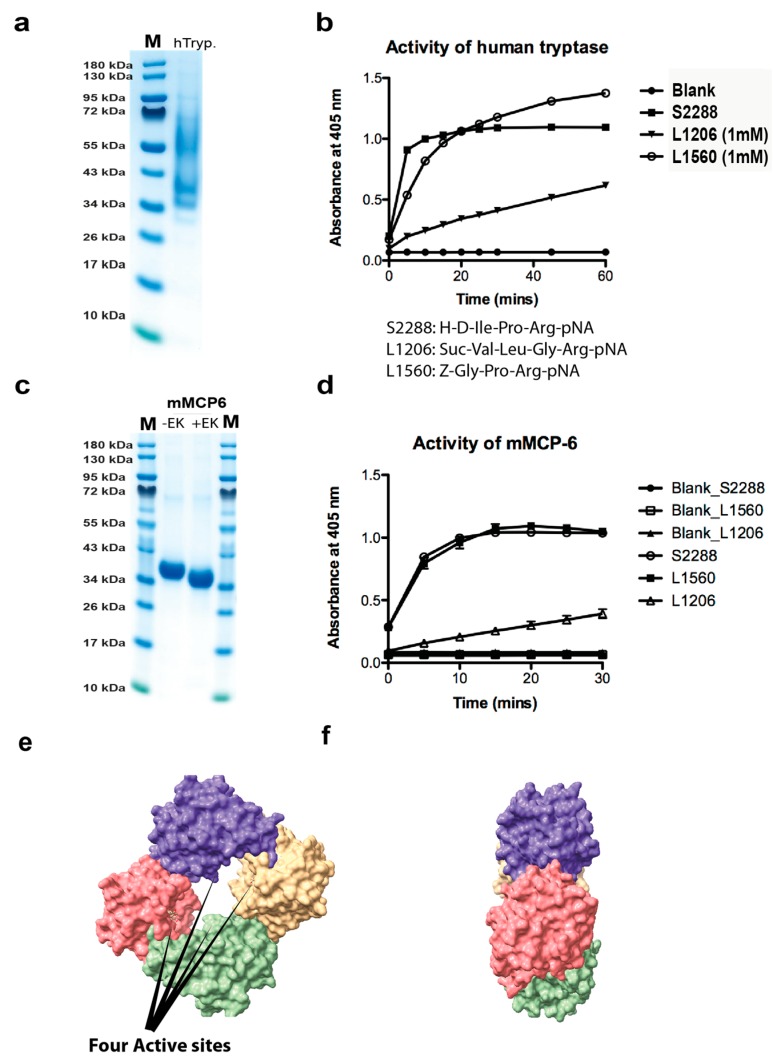
Analysis of the recombinant human and mouse mast cell tryptases used for the substrate analysis by SDS-PAGE and chromogenic substrate assay. (**a**) Approximately two micrograms of the recombinant human tryptase was separated on a 4–12% gradient SDS-PAGE gel and stained with colloidal Coomassie blue solution. Several bands starting at approximately 36 kDa up to almost 70 kDa were seen, indicating heterogenous glycosylation. (**b**) The activity of the human recombinant tryptase was assayed against three different chromogenic substrates. All three are substrates for tryptic enzymes due to the Arg in the P1 position. The human tryptase showed good activity against all three, but the best activity against the two substrates occurred when the Arg was preceded by a Pro residue. (**c**) The purified recombinant mouse mast cell tryptase (mMCP-6) before and after enterokinase cleavage. (**d**) The activity of the mouse recombinant tryptase was assayed against three different chromogenic substrates. The substrate preference for the mouse enzyme was almost identical to its human counterpart shown in (**b**). (**e**,**f**) Three-dimensional structural models of the human tryptase tetramer (PDB No: 2fs9). The space filling model of the human MC tryptase is shown from two angles, from the front looking into the tetramer and from the side [20]. The four active sites in the middle of the tetramer are marked by four arrows. The pictures were visualized in the UCSF Chimera program.

**Figure 2 ijms-20-05147-f002:**
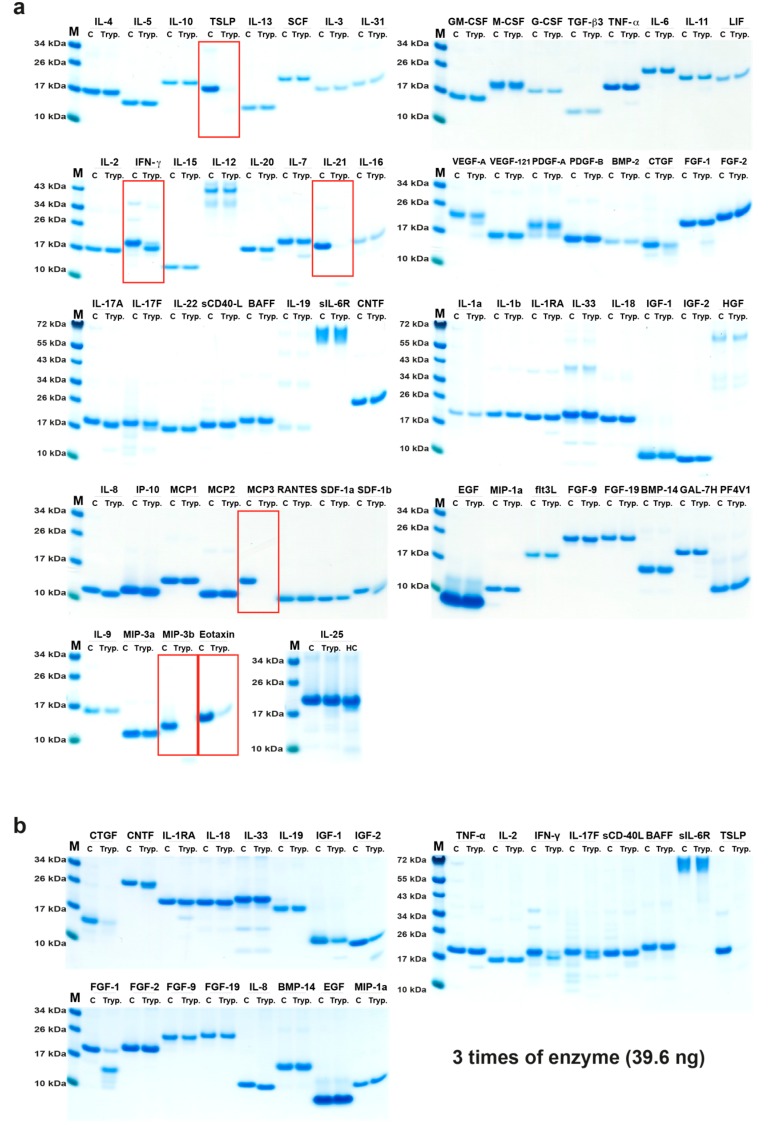
Cleavage analysis of a panel of 69 recombinant human cytokines and chemokines. The various recombinant cytokines and chemokines were divided into two separate tubes. One was kept as negative control (C) where PBS (no enzyme) was added and one was cleaved with the human tryptase. The cleavage was performed at 37 °C in PBS buffered solution (pH 7.3) for 2.5 h. The samples were separated on 4–12% SDS-PAGE gels under reducing conditions. Size markers are found at the left side of each gel. The gels were stained in colloidal Coomassie blue solution. (**a**) Cleavage with 13 ng of tryptase. (**b**) Cleavage using three times higher enzyme concentration. The cytokines and chemokines that were efficiently cleaved or trimmed are marked by red rectangles.

**Figure 3 ijms-20-05147-f003:**
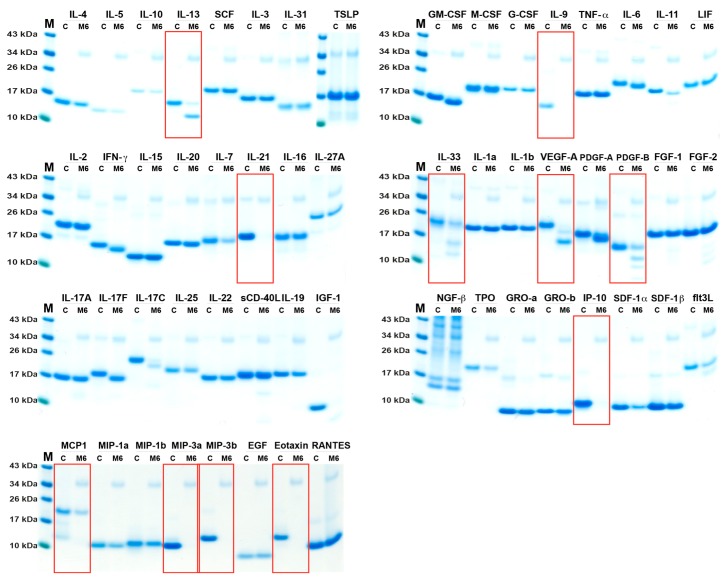
Cleavage analysis of a panel of 56 recombinant mouse cytokines and chemokines. The various recombinant cytokines and chemokines in buffer solution were divided into two separate tubes. One was kept as negative control (C) where PBS (no enzyme) was added and one was cleaved with the mouse tryptase (mMCP-6). The cleavage was performed at 37 °C in PBS at pH 7.3 for 2.5 h. The samples were separated on 4–12% SDS-PAGE gels under reducing conditions. Size markers are found at the left side of each gel. The gels were stained in colloidal Coomassie blue solution. The cytokines and chemokines that were efficiently cleaved or trimmed are marked by red rectangles.

**Figure 4 ijms-20-05147-f004:**
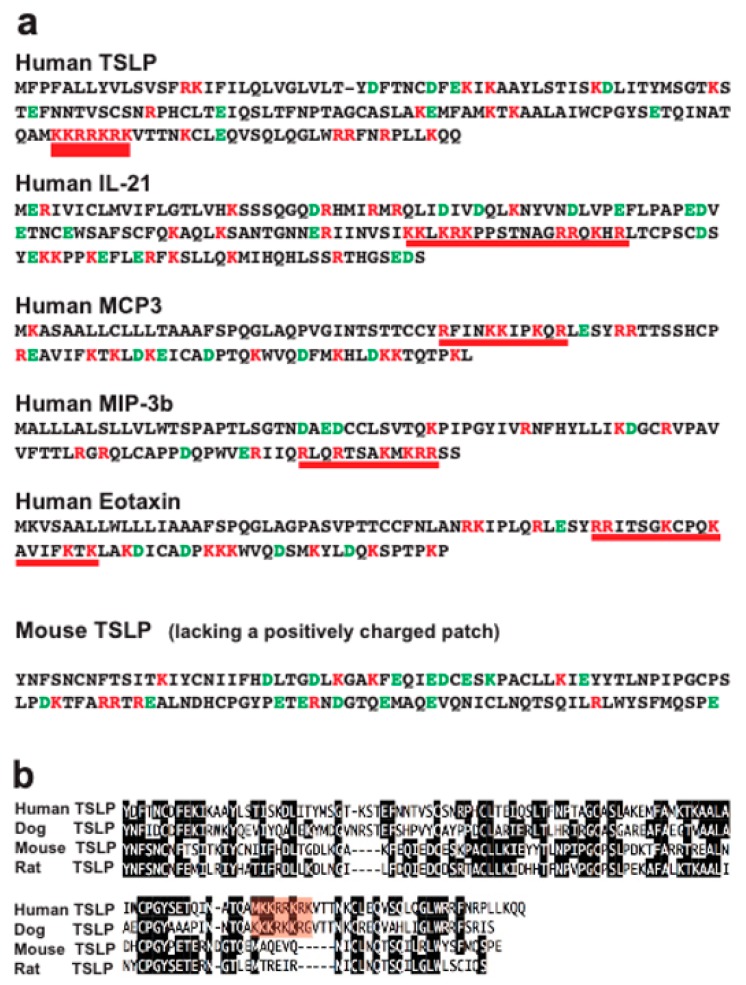
(**a**) Analysis of the primary amino acid sequences of human TSLP, IL-21, MCP3, MIP-1b, eotaxin, and mouse TSLP. The amino acid sequences of the five cytokines and chemokines that were efficiently cleaved by the human skin tryptase are shown in one letter code. Negatively charged residues are marked in green and positively charged in red. The regions that are highly positively charged and that may act as targets for the tryptase are marked by a red thick line. In (**b**) an alignment of human, dog, mouse, and rat TSLP is presented. The conserved residues between the majority of the four sequences are marked in black. The positively charged patch found in human and dog TSLP is marked in red for more easy identification.

**Figure 5 ijms-20-05147-f005:**
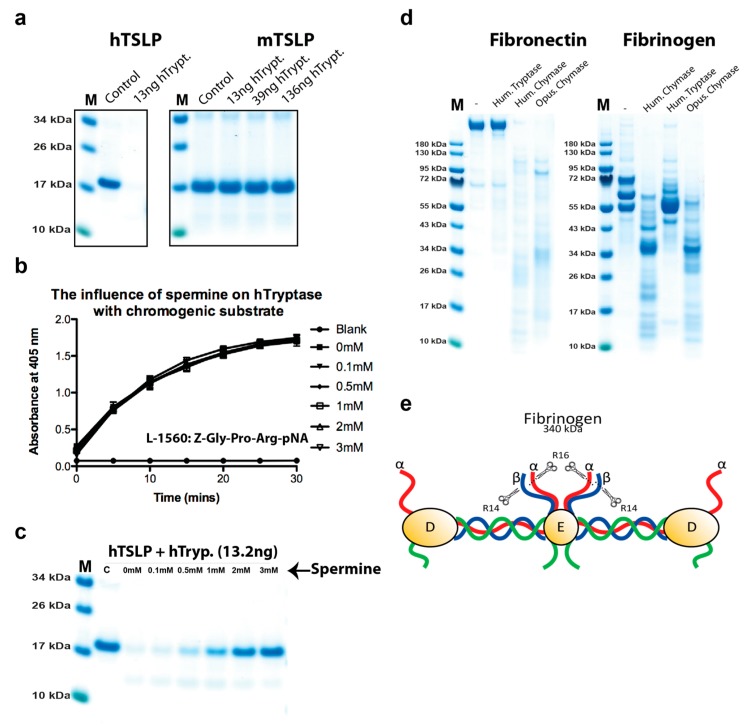
Cleavage analysis of human and mouse TSLP, the analysis of the effect by spermine on tryptase activity, and the cleavage of human fibrinogen and fibronectin by human tryptase, HC, and opossum chymase. In (**a**), 2.4 micrograms of active human TSLP was divided into two tubes and in one of the tubes human tryptase was added. The two tubes were incubated at 37 °C for 2.5 h. Following addition of sample buffer and beta mercaptoethanol, the samples were separated on a 4–12% SDS PAGE gel. A similar analysis was performed on mouse TSLP, which lacks the positively charged patch that is present in human and dog TSLP (Figure 4). In (**b**,**c**), the effect of spermine on cleavage by the human tryptase was analyzed by adding spermine to the cleavage reaction at different concentrations ranging from 0.1 to 3 mM and tested against both a chromogenic substrate (**b**) and a macromolecule as represented by human TSLP (**c**). As can be seen from the figure, spermine had no effect on the cleavage of the chromogenic substrate, but a potent effect on the cleavage of human TSLP. In (**d**), the cleavage of human fibrinogen and fibronectin by human tryptase, HC, and opossum chymase have been analyzed. Purified human fibrinogen and fibronectin were cleaved with three different enzymes, human tryptase, human chymase, and opossum chymase. In (**d**), we can see that no cleavage could be detected with the tryptase on fibronectin, but a very potent effect on this target by both human and opossum chymase. We can also see that human tryptase trims the ends of fibrinogen, whereas both human and opossum chymase has a much more pronounced effect on this target by also cleaving into the more tightly folded parts of fibrinogen. In (**e**), a schematic presentation of the overall structure of fibrinogen is presented [25]. The small scissors and R14 and R16 represent the cleavage sites in these regions of the α and β chains of fibrinogen by the coagulation enzyme thrombin.

**Figure 6 ijms-20-05147-f006:**
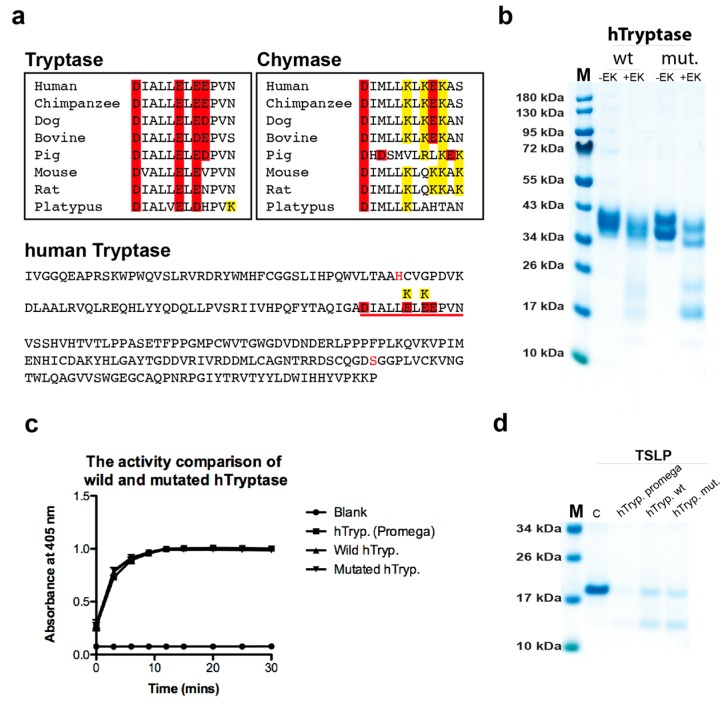
The effect on cleavage by mutating two negatively charged residues close to the active site of human tryptase. By analysis of the primary structure of the human tryptase we have observed a marked difference concerning charge close to the active site between tryptases and chymases. C-terminally of the asparagine residue of the catalytic triad all tryptases have a negative patch of three negatively charged residues (**a**). This region including the Asp of the catalytic triad and the three negatively charged residues is underlined in red in the sequence of the entire human tryptase shown in the bottom panel of (**a**). In this region, all chymases instead have two positively charged residues and one negatively charged amino acid (**a**). Could this region be involved in target selection? Two residues that differ between tryptases and chymases in the human tryptase was therefore mutated to study their involvement on cleavage (**a**). The wt and mutant enzymes were produced in HK-293-EBNA cells and activated by enterokinase cleavage, lowering the pH to 6.0, and adding heparin. (**b**) SDS-PAGE gel analysis of wt and mutant enzyme before and after enterokinase cleavage. (**c**) Cleavage of chromogenic substrates and (**d**) the cleavage of human TSLP by the pichia produced enzyme and the wt and mutated tryptase produced in mammalian cells.

**Figure 7 ijms-20-05147-f007:**
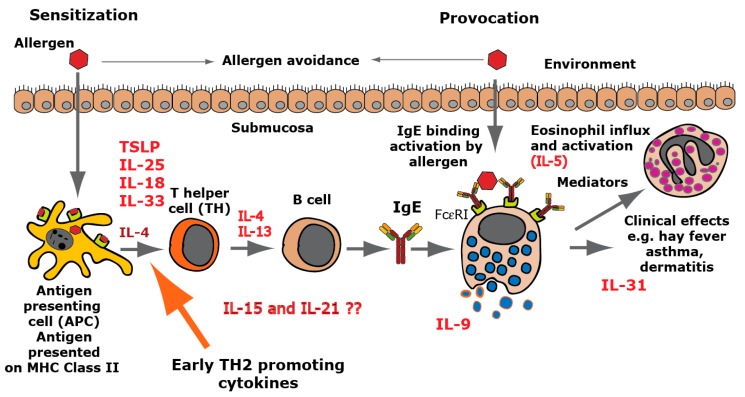
TH2-inducing cytokines in allergy development. A number of cytokines have been shown to be potent inducers of TH2-mediated immunity. The most well characterized are thymic stromal lymphopoietin (TSLP), IL-33, IL-18, IL-25, and IL-4. IL-18 has been shown to be a potent inducer of TH2-mediated immunity when present alone and not in combination with IL-12. Interestingly when present together with IL-12, IL-18 acts instead as an enhancer of TH1-mediated immunity [40]. IL-4 and IL-13 are the only two cytokines known to induce isotype switching in B cells to IgE [41]. IL-5 is important for eosinophil infiltration activation and proliferation, and IL-31 acts as an inducer of itch in skin with atopic dermatitis. IL-9 is, in mice, an inducer of mucosal mast cell differentiation [42]. Both IL-15 and IL-21 have been found to have TH2-promoting activity as described in the text. Cleavage of the TH2-initiating early cytokines would most likely result in a dampening effect on TH2-mediated immunity.

**Table 1 ijms-20-05147-t001:** Summary of the results from the cleavage analysis of cytokines and chemokines by the human and mouse tryptases and the hMC chymase (HC).

Cytokine	hTryp.	M6	HC	Cytokine	hTryp.	M6	HC
IL-4	−	−	−	IL-1α	−	−	
IL-5	−	−	−	IL-1β	−	−	−
IL-10	−	−	−	IL-1RA	+		
TSLP	++++	−	+	IL-33	−	+	++++
IL-13	−	+++	+	IL-18	−		+++
SCF	−	−	−	IGF-1	−	++++	
IL-3	−	−	+	IGF-2	−		
IL-31	−	−	−	HGF	−		
GM-CSF	−	+	−	IL-8	++		−
M-CSF	−	−		IP-10	−	++++	−
G-CSF	−	−	−	MCP-1	−	+	−
TGF- β3	−			MCP-2	−		−
TNF-α	−	−	−	MCP-3	++++		−
IL-6	−	+	++	RANTES	−	−	−
IL-11	−	+	+	SDF-1α	−	++	−
LIF	−	−	+	SDF-1β	+	−	−
IL-2	−	−	−	EGF	−	−	−
IFN-γ	+++	+	+	MIP-1α	−	−	-
IL-15	−	−	++++	flt3L	−	+	++
IL-12	−		−	FGF-9	−		
IL-20	+	−	−	FGF-19	−		
IL-7	+	+	−	BMP-14	−		
IL-21	++++	++++	−	GAL-7H	−		
IL-16	−	−		pF4V1	−		
VEGF-A	+	++++	−	IL-25	−	−	−
VEGF-121	−			IL-9	−	++++	
PDGF-A	−	+	−	MIP-3a	−	++++	
PDGF-B	−	+++	−	MIP-3b	++++	++++	
BMP-2	−			Eotaxin	+++	++++	
CTGF	++		+	IL-27A		-	
FGF-1	+	−	+	IL17C		++++	
FGF-2	−	+	+	NGF-β		−	
IL-17A	+	−	−	TPO		+	
IL-17F	+	+	−	GRO-a		−	
IL-22	−	−	−	GRO-b		−	
CD40L	−	+	−	MIP-1b		−	
BAFF	−		+				
IL-19	−	−	−				
sIL-6R	−						
CNTF	−						

All the results concerning the MC chymase originate from an earlier study [17]. The analysis is based on the SDS-PAGE analysis in Figure 2 and Figure 3. (−) no cleavage activity was observed. (+) showed minor activity, (++) and (+++) partial cleavage. (++++) complete or almost complete cleavage by the enzyme.

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
