# Peer review of "Highly Selective Cleavage of TH2-Promoting Cytokines by the Human and the Mouse Mast Cell Tryptases, Indicating a Potent Negative Feedback Loop on TH2 Immunity"

_ijms, 2019, doi:10.3390/ijms20205147_

Round 1

Reviewer 1 Report

The authors characterize mast cell tryptases in detail and therefore define their role in the immune system. This work extends their previous study on mast cell chymases and cathepsin G. The work is done very carefully with all the necessary controls. I particularly appreciate the final scheme (Figure 7) that nicely illustrates the role of mast cell and especially mast cell secreted proteases in the regulation of the immune system.

The article is written very clearly, but it is perhaps unnecessarily long and the shortening of some parts would be beneficial for the readers.

I can only mentioned few things that could help improve the work.                           

Minor revision - since a large number of potential substrates have been studied, those that have been cleaved by the tested enzymes are not clearly visible in the images, namely in figure 2 and 3. For better understanding of the article it would certainly be helpful to mark the lines with cleaved products  (color, rectangle, arrow, ...), so that the reader can more easily find them. Alo shorteng of the article would be beneficial.

Author Response

All the cleaved cytokines and chemokines in figures 2 and 3 have been marked with red boxes.

We have done our best to shorten the manuscript by deleting parts of the last part of the introduction however we felt it difficult to shorten results and discussion as it would make the study less readable-Sorry.

Reviewer 2 Report

Fu et al. have analyzed the activity of MC tryptase on cytokines and chemokines in human and mouse. Both enzymes showed a substrate specificity, and only a limited number of cytokines and chemokines were cleaved by the enzymes. The experiments were well conducted, and the results were clear. It is interesting that TH2-promoting inflammatory cytokines were selectively cleaved by these enzymes. However, since all experiments were done in vitro by using recombinant proteins, the present results are too preliminary to describe their functions in vivo. In addition, molecular basis underlying the substrate specificity has not been fully explored. Therefore, the present manuscript needs a certain amount of work for acceptance.

Major concerns:

The primary amino acid sequences of the uncleaved cytokines/chemokines should be examined. It is very important to examine whether or not the uncleaved cytokines/chemokines have the KKRRKRK or related basis amino acid sequences in their primary amino acid sequence. It is possible that the existence of the KKRRKRK or the related sequence is necessary, but not sufficient, for the specific cleavage by the tryptases. The requirement of higher order protein structure might be considerable for the substrate specificity. It would be very nice to add new column for the existence of the KKRRKRK or the related sequence in Table 1.

How the authors determine the condition of enzyme reaction, such as the amounts of enzymes and substrates? Is it similar to those in vivo?

In the SDS-PAGE analyses, the cleaved protein fragments were not appeared on the gel. The smaller protein fragments might be detectable for TH2-promoting cytokines in different experimental conditions.

In all figures A, B and C….. should be corrected as (a), (b) and (c)…. as in a template file. In Figure legends, “panel” should be deleted.

Specific comments:

The statement in the title“indicating a potent negative feedback loop on TH2 immunity” is too speculative.

Page 2, line 25, The statements that “the tryptase found in human MCs is a homo or heterotetramer of three closely related proteases the b1, b2 and b3 tryptases” may not be correct. It is a tetramer of 4 major tryptase isoforms, alpha, b1, b2 and b3. The alpha isoform is hypomorphic, but it does form as a partner of b1 or b2. (Trivedi et al., J. Allergy Clin Immunol. 2009;124:1099-105)

Figure1B and 1D: The activity of L1206 was lower than S2288 and L1560. An additional explanation is needed.

Page 2: The last paragraph of the introduction is too long and overlapping with the results.

Page 2: The last sentence of the introduction: “his” should be corrected as “this”.

In Figure 1B, S-2288, L-1206 and L-1560 should be corrected as S2288, L1260 and L1560.

Figure 1E and 1F Did the authors determine the high order protein structure by themselves? If so, the methods should be added. If not, how did the authors make this model?

Figure 5E It is not the authors’ original data and should be deleted.

Figure 7 is a nice drawing, but is beyond the authors’ results obtained from in vitro study. It should be deleted at present.

A reference should be sited for the followings.

・Page 2, line 16: Numerous examples on the role of these enzymes in the degradation of inflammatory mediators have been described, indicating that this may be one of the important functions of these enzymes.

・Page 2, line 28: Interestingly the active sites of these four subunits are positioned in the center of the tetramer making them less accessible for larger substrates.

・Page 4, line 10:

To confirm the initial results, the experiment was repeated under the same conditions as previously described.

・Page 9, line 10:

Several proteins have previously been shown to be trimmed or degraded by human tryptase including fibrinogen and fibronectin.

・Page 9, line 13:

It is known that the N terminal tails of fibrinogen α and β chains are relatively unstructured and open for cleavage by thrombin (Fig.5E).

Author Response

Reviewer 2

We have analyzed the sequence of all the non-cleaved cytokines and chemokines to look for positively charged patches. Only three had such patches, VEGF-A, PDGF-A and B and one had a minor such patch in the C-terminal of the protein and that was IFN-gamma. IFN gamma is also trimmed in the end probably the C-terminal. Non of the others had any strong positive patch and non of them had the same sequence as human TSLP, KKRRKRK. Screening the entire human proteome only identified two additional proteins having that sequence, one seven transmembrane receptor and one large protein of unknown function. No cytokine or chemokine of the entire human proteome had that sequence. It is most likely as the reviewer correctly say that also the higher order structure is of importance. Interestingly, the mouse tryptase mMCP-6 cleaves all the three cytokines with a positive patch identified by this new analysis, VEGF-A, PDGF-A and B indicating that small differences in higher order structure of the target and /or the protein can have a major impact on the cleavage. We have now added a section describing these findings in the result section.

Physiological concentration: Very many possible concentrations may be physiologic as close to the cell the enzyme concentration is very high, a little bit away the concentration is lower and at a large distance most likely not active due to presence of protease inhibitors so we tried to obtain a picture as clear as possible of the activity of these proteases on these cytokines and chemokines and therefore chose a concentration where the cytokines and chemokines are clearly visible on the gels and that the enzyme concentration was adjusted to its activity- less of highly active and more of enzymes with low activity.

Very small peptides are very difficult to show on SDS-PAGE gels as they often run with the salt front and also do not get fixated in the gel, which is needed for staining. In some cases we see bands indicating only partial cleavage. However very small fragments needs to be analyzed by MS or HPLC.

We have now corrected all figures and figure legends to (a), (b) and (c)…..

Speculative: We clearly say that it indicates which I think is correct. We were really struck by the results when analyzing the tryptase. The published study of the human mast cell chymase published in J.I. in 2017 gave a strong indication for a role of mast cell proteases in regulating TH2 immunity and then when we analyzed the tryptase all the missing TH2 cytokines from the chymase cleavage study were identified as prime targets for the tryptase. We felt that it was almost too striking to be true. However, the study was performed totally unbiased why we feel that indicated is almost too weak statement. As stated in the discussion there is also a number of in vivo studies that support our findings.

Correctly stated by the reviewer the beta tryptase can also combine with the alpha tryptase that so nicely has recently been shown also to affect target selectivity and stability by the lab of Larry Schwartz. We have now added that info and the reference to the introduction.

The only explanation for the difference in activity on these chromogenic substrates is the substrate specificity of the tryptase. The tryptase seems to prefer Pro-Arg over Gly-Arg in the peptides, possibly due to stronger binding of the larger Pro compared to the small amino acid Gly. A factor that could be due to the use of short peptides without amino acids at the C terminal side of the cleavage site, as always is the case when using this type of chromogenic substrates.

We have now reduced the last part of the introduction.

S-2288 L-1206 and L-1560 have been corrected.

The high ordered structures are based on published work and 3D structure open resource and we have made the figures our-selves. These are there to make it easier for a reader to follow the reason in the text. I myself always want such explanatory figures in articles not to need to spend hours searching in earlier published work to understand the reasoning in an article. They are just there to make the article readable for a larger audience. We have now also described the source of the 3 D structure in the figure legend and added a reference for the initial 3D structural studies.

 This is also the reason why we include figure 7, deleting this figure would result in an article more difficult for the majority of readers to follow. We have also made this figure in Adobe Illustrator ourselves. This figure is just there to give a summary of what is known about various cytokines involved in a TH2 mediated immune response and cytokines affecting cells involved in the development of an atopic, IgE mediated allergy. I think the article should be much less readable and more difficult to follow if we deleted these figures. The figures are also all made by us so it is no copies of artwork by others.

References

Page 2 line 16 Numerous examples…….. Almost the entire page before this sentence is full of references to this statement involving a large number of cytokines and chemokines.

The active site in the center is based on the 3D structure that now is deposited at the internet structural database which we have used for designing the Figure 1 E and F. We have also added a refence to the original 3D structure of the beta tryptase.

Repeated. This is just to state that the experiment has been repeated and we feel it not necessary to show both experiment 1 and experiment 2 which both showed the same thing they were almost identical. To clarify that this is just a confirmatory experiment we now state (data not shown).

We have now added references to the fibronectin and fibrinogen cleavage and the unstructured ends of fibrinogen.

We hope these corrections and changes to the manuscript make it acceptable for publication in IJMS.

Round 2

Reviewer 2 Report

The authors have addressed most of the comments raised by this reviewer. As the quality of the manuscript is improved, I recommend it for publication.